# MVGSR: Multi-View Consistency Gaussian Splatting for Robust Surface Reconstruction

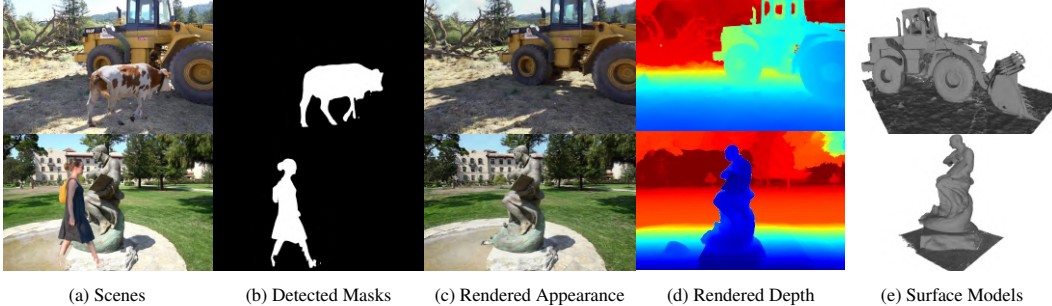

| (a) Scenes | (b) Detected Masks | (c) Rendered Appearance | (d) Rendered Depth | (e) Surface Models |

Figure 1: Given input images with distractors (a), our MVGSR leverages multi-view feature contrast to generate accurate distractor masks (b). This effectively suppresses gradient leakage and enables high-quality surface reconstruction with photorealistic rendering fidelity (c-d).

## Abstract

3D Gaussian Splatting (3DGS) has recently emerged as a powerful approach for high-quality dense surface reconstruction of unknown scenes. However, existing methods are limited by the assumption of static environments. In practice, they often fail in everyday scenarios with dynamic objects and transient distractors that resulted in floating artifacts, geometric distortions, and view-dependent appearance errors in 3D reconstructed models. We propose a robust surface reconstruction framework that leverages Gaussian models together with a heuristics-guided distractor masking strategy. Unlike prior methods that rely on MLP-based uncertainty modeling for distractor segmentation, our approach uses multi-view feature consistency to separate distractors from static content. This allows us to obtain precise distractor masks in the early stage of training. To further improve reconstruction, we introduce a pruning mechanism that evaluates the visibility of each Gaussian across views. Specifically, it resets the transmittance of unreliable points and thus suppresses floating artifacts to yield a more compact representation while preserving rendering quality. Finally, we design a multi-view consistency loss that enforces both structural and color coherence across views to improve the fidelity of Gaussian splats in distractor-heavy scenes. Extensive experiments demonstrate that our method achieves state-of-the-art geometric accuracy and rendering fidelity while remaining robust in dynamic and cluttered environments. The code will be made publicly available on paper acceptance.

## 1 Introduction

Novel view synthesis and geometry reconstruction are fundamental yet challenging tasks in computer vision. They play a central role in augmented and virtual reality (AR/VR), autonomous driving, and digital twins. Recent years have witnessed the rise of neural implicit representations (Mildenhall et al., 2021; Rosinol et al., 2023; Di et al., 2024), which model 3D scenes with remarkable accuracy and generalizability. More recently, 3D Gaussian-based representations (Kerbl et al., 2023; Li et al., 2024; Huang et al., 2024) have emerged as a promising alternative. Specifically, they offer faster processing and superior rendering quality compared to implicit neural fields.

Despite their impressive performance, both NeRFs and 3DGS assume a static scene and aim to recover consistent geometry while capturing non-Lambertian effects. This assumption is rarely satisfied in real-world scenarios where transient distractors are unavoidable. Distractors may appear as occlusions, *e.g.* moving pedestrians or vehicles, and illumination artifacts, *e.g.* dynamic shadows cast by the photographer. As illustrated in Fig. 1, such distractors violate the static-scene assumption and lead to incorrect surface geometry and artifacts clustered near the camera lens. Moreover, distractors that appear inconsistently across viewpoints are often misinterpreted as non-Lambertian effects. This misinterpretation produces false viewpoint-dependent colors on object surfaces.

To improve novel view rendering performance in scenes containing distractors, prior work has developed two main paradigms. The first is *uncertainty modeling*, which trains a Multi-Layer Perceptron (MLP) to capture discrepancies between rendered and ground-truth images in a semantic feature space, *e.g.* DINO. The predicted uncertainty is then used to generate distractor masks (Ren et al., 2024; Yu et al., 2024; Sabour et al., 2024). Although effective in some cases, these methods introduce additional degrees of freedom. This complicates loss tuning and requires careful trade-offs for Pareto-optimal objectives (Sabour et al., 2023). The second paradigm is *residual decomposition*, which isolates distractors by analyzing the color residuals between the rendered and ground-truth images (Sabour et al., 2023). For example, RobustNeRF identifies distractors as outliers by iteratively thresholding residuals. Desplat (Wang et al., 2025) and HybridGS (Jingyu Lin, 2025) extend this line by modeling distractors with 2DGS. However, such methods frequently misinterpret the underfitting of high-frequency details during early training as distractor noise. This misidentification causes gradient leakage that leads to artifacts and degraded geometry in the distraction-free regions (Ren et al., 2024). Although these approaches alleviate the artifacts in rendered RGB images, they still overlook the surface quality of the reconstructed models. We observe that Gaussians in distractor regions deviate from the surface at early stages; as training proceeds, color residuals shrink while depth error accumulates, leaving off-surface Gaussians hard to remove and causing persistent artifacts or incorrect geometry.

In contrast to these approaches, the proposed method focuses on distractor-free surface reconstruction, and our key insight is that ***distractors appearing in only a few images lack semantic-geometry consistency across multiple views***. This results in noticeable differences in the features extracted by pre-trained models. We make use of this property to separate distractors from static scene content in the early stages of training. Compared with iterative uncertainty-based masks, multi-view comparison offers two benefits. First, it avoids confusing distractors with fine details, and thus preserving high-quality surface reconstruction. Second, it imposes strict mask boundaries to block gradient leakage during Gaussian splitting. As a result, artifacts and viewpoint-dependent color errors are significantly reduced. To further address floating artifacts and ghosting from uncertain geometry, we introduce a pruning measure based on multi-view contributions. By resetting the transmittance of occluding objects, we mitigate gradient flow issues caused by occlusion. This reduces floating artifacts, compresses the point cloud representation, and maintains rendering quality with only minimal degradation. Finally, we incorporate a multi-view consistency loss that enforces both structural and color coherence across views. This improves the reconstruction fidelity of Gaussian splats, even in challenging distractor-heavy scenes. Our contributions are summarized as follows:

- We propose a semantic-geometry consistency strategy that detects distractors before surface reconstruction, enabling strict distractor masking and significantly reducing floating artifacts.

- We introduce a Gaussian pruning strategy based on multi-view contributions that removes floating artifacts with negligible loss of rendering capability.

- We design a multi-view consistency loss that enforces structural and color alignment across viewpoints, yielding high-fidelity rendering and accurate surface models in scenes with distractors.

- We provide both real-world and synthetic datasets to facilitate the evaluation of depth estimation and surface reconstruction in scenes with distractors.

## 2   RELATED WORK

**Neural/GS-based Surface Reconstruction.** NeRF-based surface reconstruction methods (Mildenhall et al., 2021; Barron et al., 2022) use 5D ray sampling to predict implicit density and color fields. These approaches achieve photorealistic rendering, but often suffer from geometric inaccuracies and high computational cost as a result of their reliance on MLP architectures. Subsequent methods

improve geometric precision using signed distance functions (SDF) such as NeuS (Wang et al., 2021), BakedSDF (Yariv et al., 2023), and UNISURF (Oechsle et al., 2021). Other works including Nerf2Mesh (Tang et al., 2023) introduce iterative mesh refinement based on rendering error feedback.

On the other hand, Gaussian splatting (GS) methods optimize explicit point-based radiance fields with various geometric priors. SuGaR (Guédon & Lepetit, 2024) constrains Gaussians to surfaces using regularization, and 2DGS (Huang et al., 2024) collapses volumetric Gaussians into view-consistent 2D representations. GOF (Yu et al., 2024) extracts geometry directly from opacity fields, and PGSR (Chen et al., 2024) assumes planar surfaces with depth recalibration to enforce multi-view consistency. Although these approaches reduce rendering cost, they often struggle with stable depth estimation and smooth surface reconstruction.

Our approach builds on 3DGS and overcomes the limitations mentioned above by progressively transforming ellipsoidal Gaussians into surfels during densification. This strategy produces smoother surfaces, improves geometric stability, and generalizes better to complex object shapes compared to prior GS methods.

**Distractor Removal.** The removal of distractors is a critical challenge for sure reconstruction in non-static scenes. Existing methods fall into two main categories: segmentation-based approaches and heuristics-based approaches.

Segmentation-based methods rely on semantic or instance segmentation models to detect objects with dynamic characteristics or to identify static regions. For example, DynaMoN (Karaoglu et al., 2023) integrates semantic maps with camera localization to enable dynamic scene reconstruction in NeRF. These methods benefit from strong priors in pre-trained models but depend heavily on the accuracy and generalization of the segmentation network.

Heuristics-based methods instead exploit cues from multi-view geometry to separate static and dynamic components. NeRF-W (Martin-Brualla et al., 2021) assumes that transient objects are typically small and treats them as residual components during training. RobustNeRF (Sabour et al., 2023) identifies transient regions using photometric residuals between the rendered and observed images.

In contrast to these approaches, our method does not rely solely on pre-trained segmentation models or photometric residuals. Instead, we leverage multi-view feature consistency to identify distractors that lack semantic coherence across views. This enables precise masking at an early stage of training and avoids the ambiguity between distractors and fine-scene details that often limits prior methods.

## 3 PRELIMINARIES OF GAUSSIAN SPLATTING

**Primitives.** 3D Gaussian Splatting (3DGS) (Kerbl et al., 2023) models a scene using a set of 3D Gaussians $\{\mathcal{G}_i\}$. Each Gaussian is defined as, $\mathcal{G}_i(\boldsymbol{x} \mid \boldsymbol{\mu}_i, \boldsymbol{\Sigma}_i) = e^{-\frac{1}{2}(\boldsymbol{x}-\boldsymbol{\mu}_i)^\top \boldsymbol{\Sigma}_i^{-1}(\boldsymbol{x}-\boldsymbol{\mu}_i)}$, where $\boldsymbol{\mu}_i \in \mathbb{R}^3$ is the Gaussian center, and $\boldsymbol{\Sigma}_i \in \mathbb{R}^{3\times3}$ is the covariance matrix. The covariance $\boldsymbol{\Sigma}_i$ is decomposed into a scaling matrix $\boldsymbol{S}_i \in \mathbb{R}^{3\times3}$ and a rotation matrix $\boldsymbol{R}_i \in SO(3)$ via $\boldsymbol{\Sigma}_i = \boldsymbol{R}_i \boldsymbol{S}_i \boldsymbol{S}_i^\top \boldsymbol{R}_i^\top$.

**Projection and Rendering.** Each Gaussian is projected into the camera coordinates using the transformation matrix $\boldsymbol{W}$ and the intrinsic matrix $\boldsymbol{K}$:

$$\boldsymbol{\mu}_i' = \boldsymbol{K}\boldsymbol{W}[\boldsymbol{\mu}_i, 1]^\top, \quad \boldsymbol{\Sigma}_i' = \boldsymbol{J}\boldsymbol{W}\boldsymbol{\Sigma}_i\boldsymbol{W}^\top\boldsymbol{J}^\top, \tag{1}$$

where $\boldsymbol{J}$ is the Jacobian of the affine approximation of the projection. The color $\boldsymbol{C}$ of a pixel $\boldsymbol{u}$ is rendered through $\alpha$-blending:

$$\boldsymbol{C} = \sum_{i \in N} T_i \alpha_i \boldsymbol{c}_i, \quad T_i = \prod_{j=1}^{i-1}(1 - \alpha_j), \tag{2}$$

where $\alpha_i$ is obtained by evaluating $\mathcal{G}_i(\boldsymbol{u} \mid \boldsymbol{\mu}_i', \boldsymbol{\Sigma}_i')$ and multiplying by a learnable opacity. The view-dependent color $\boldsymbol{c}_i$ is represented using spherical harmonics (SH). $T_i$ denotes the accumulated transmittance, and $N$ is the number of Gaussians intersected by the ray. The Gaussian center can be explicitly transformed into camera coordinates as:

$$[x_i, y_i, z_i, 1]^\top = [\boldsymbol{W} \mid \boldsymbol{t}][\boldsymbol{\mu}_i, 1]^\top. \tag{3}$$

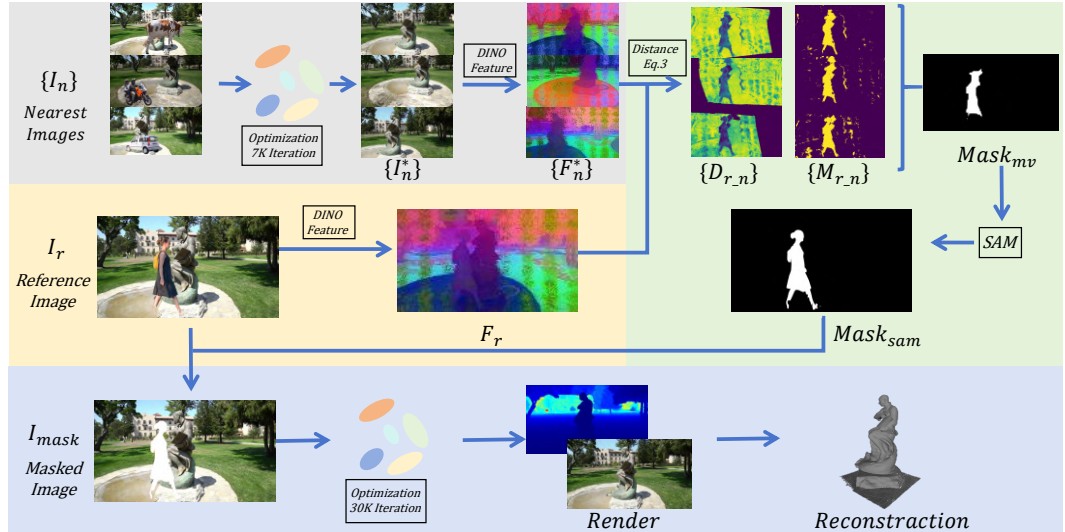

Figure 2: The detailed architecture of our MVGSR Framework. Images with distractors are fed to the system that makes use of multi-view consistency Gaussian Splatting algorithm to achieve robust surface reconstruction for non-static environments.

**Depth and Normal Rendering.** For each Gaussian, the direction of its smallest scale factor corresponds to its surface normal $\boldsymbol{n}_i$. Similar to (Jiang et al., 2023; Cheng et al., 2024), the depth and normal maps under the current viewpoint are obtained through $\alpha$-blending:

$$\boldsymbol{D} = \sum_{i \in N} T_i \alpha_i z_i, \quad \boldsymbol{N} = \sum_{i \in N} T_i \alpha_i \boldsymbol{R}_c^\top \boldsymbol{n}_i, \tag{4}$$

where $\boldsymbol{R_c}$ is the rotation from the camera to the global coordinate system.

## 4 OUR METHODOLOGY

Given a set of images of a scene containing outliers, our goal is to achieve high-precision surface reconstruction while maintaining robust view rendering. Prior works on outlier masking (Sabour et al., 2023; 2024) are generally based on photometric errors to estimate masks. However, such approaches cannot reliably distinguish true outliers from fine reconstruction details and thus often lead to inaccuracies. To address this limitation, we propose a method that leverages feature similarity across multiple views to detect outliers. Specifically, we first extract the features using a self-supervised 2D foundation model (Oquab et al., 2023). We then obtain a rough scene representation and compute mappings between a reference view and its adjacent views with only a few training iterations. Finally, we derive an initial mask that highlights potential outliers by comparing feature similarities.

We further refine this mask to improve boundary accuracy. The points sampled from the initial mask are passed through a lightweight segmentation model to obtain a more precise mask. Retraining with this refined mask mitigates gradient leakage during optimization and preserves reconstruction quality. In addition, we introduce a pruning mechanism based on cross-view visibility. This step resets the transmittance of unreliable points to effectively remove artifacts, thus yielding a more compact and reliable representation. The procedure for obtaining the initial mask is described in Sec. 4.1. Multi-view pruning is presented in Sec. 4.2, and the loss function used for optimization is discussed in Sec. 4.3.

### 4.1 SEMANTIC-GEOMETRY CONSISTENCY FOR DISTRACTOR DETECTION

Similar to traditional 3D Gaussian Splatting methods (Kerbl et al., 2023; Li et al., 2024), scenes with distractors are first processed in an initialization stage. This stage uses sparse point clouds and camera parameters generated from structure-from-motion (SfM). Based on this setup, we perform a

rough reconstruction and obtain the initial depth map $D$, surface normals $\mathbf{N}$, and a rendered image $I^*$. These outputs provide a baseline representation of the scene, which is refined in later stages to improve accuracy and handle distractors.

Our distractor detection strategy consists of three steps.

**Step 1: Multi-view feature association.** We associate Gaussian surfels in the reference view $I_r$ with the corresponding surfels in a neighboring view $I_n$. This association is based on the relative camera pose $\mathbf{R}, \mathbf{t}$. For a pixel $\mathbf{p}_r$ in $I_r$, the corresponding pixel $\mathbf{p}_n$ in $I_n$ is computed via $\mathbf{p}_n = \mathbf{H}_{nr}\mathbf{p}_r$, and the homography is:

$$\mathbf{H}_{nr} = \mathbf{K}_n \mathbf{R}_{nr} \left( \mathbf{I} + \frac{1}{d_r} \cdot \mathbf{t}_{rn} \mathbf{n}_r^\top \right) \mathbf{K}_r^{-1} \tag{5}$$

Here, $\mathbf{K}_r, \mathbf{K}_n$ denote the camera intrinsics, $d_r$ is the depth of pixel $\mathbf{p}_r$ in $I_r$, $\mathbf{n}_r$ is its surface normal, and $\mathbf{I}$ is the identity matrix.

Following methods (Ren et al., 2024; Kulhanek et al., 2024), we then extract the features with DINOv2 (Oquab et al., 2023) to obtain the feature maps $F_r$ and $F_n^*$. Each feature vector in has 384 channels. Instead of using the raw neighboring image $I_n$, we rely on the rendered image $I^*$ to remove transient distractors that would otherwise corrupt the feature matching (see Fig. 2). For the corresponding pixels $\mathbf{p}_r$ and $\mathbf{p}_n$, the feature vectors are interpolated from the downsampled maps $F_r$ and $F_n^*$. Since the feature maps are reduced by a factor of 14 relative to the original images, bilinear interpolation is used to retrieve accurate feature values at non-integer pixel coordinates. The feature distance is then defined as:

$$\text{distance}(\mathbf{p}_r, \mathbf{p}_n) = \text{abs}\left( \frac{F_r(\mathbf{p}_r) \cdot F_n^*(\mathbf{p}_n)}{\|F_r(\mathbf{p}_r)\| \, \|F_n^*(\mathbf{p}_n)\|} \right). \tag{6}$$

Pixels with a distance below a threshold $\delta_{near}$ are marked as distractors in a preliminary mask $M_{r_n}$.

**Step 2: Multi-view filtering.** Each preliminary mask $M_{r_n}$ may include false detections due to noise and rough geometry. To circumvent this issue, we use multi-view consistency. For each pixel labeled as clutter in $M_{r_n}$, we check its visibility in other adjacent views. It is kept in the final multi-view mask $Mask_{mv}$ if at least two adjacent views also classify the pixel as a distractor.

**Step 3: Boundary refinement.** The coarse mask $Mask_{mv}$ often has imprecise boundaries. To refine them, we use the Segment Anything Model (SAM) (Kirillov et al., 2023). We uniformly sample positive points inside the clutter mask and negative points in the background. These prompts and the reference image $I_r$ are fed into SAM to obtain the refined mask $Mask_{sam}$ with sharper boundaries:

$$Mask_{sam} = \text{SAM}(I_r, \mathcal{P}, \mathcal{N}). \tag{7}$$

## 4.2 Multi-view Pruning

In cluttered scenes, optimization frequently drives Gaussians to cluster near the camera. These resulting floaters are incorrect representations of viewpoint-dependent effects produced by clutter visible in only a few frames, and they cause the line of sight to reach the transmittance limit prematurely. This blocks gradient propagation and hinders optimization. A common solution in 3DGS is to reset the opacity of all Gaussians every few iterations. This serves as a control mechanism that reopens gradient flow and prunes Gaussians that cannot regain high opacity. However, this reset strategy is not sufficient in cluttered regions due to two issues. First, the presence of masks leads to more Gaussian splitting after each opacity reset. Second, imperfect masks amplify this problem by introducing additional floaters.

From a geometric reconstruction perspective, the spatial distribution of Gaussians is mainly governed by their centers and the per-ray opacity (alpha) that controls transmittance along viewing rays (Kerbl et al., 2023). Evaluating contribution with respect to alpha provides a geometry-aware signal and promotes higher geometric fidelity. In contrast, some existing schemes assess contribution using image-space color gradients, but prior observations indicate that gradient magnitude is confounded by the Gaussian footprint/size, which biases pruning toward removing larger yet low-opacity splats. This bias harms reconstruction in weak-texture regions where large, semi-transparent splats are essential to faithfully explain smooth surfaces. Therefore, we adopt an alpha-based, multi-view contribution metric and use it to drive pruning.

To address the limitations mentioned above, we propose *multi-view contribution-based pruning* (MV-Prune). We define the multi-view contribution of a Gaussian $p$ as:

$$\mathbf{C}_{MV}(p) = \sum_{\mathbb{V}_k \in \mathbb{V}} \left( \sum_{p \in \mathbb{V}_k} \alpha_{i(p)} \prod_{j=1}^{i(p)-1} (1 - \alpha_j) > \delta_{\text{opacity}} \right), \tag{8}$$

where $\mathbb{V}_k$ denotes a training viewpoint, and $p$ is the Gaussian under evaluation. The contribution of $p$ to viewpoint $V_k$ is determined by the cumulative transmittance along the ray that intersects $p$. When this cumulative transmittance exceeds the opacity threshold $\delta_{\text{opacity}}$, the contribution of $p$ in that view increases by one.

A Gaussian is pruned when its total contribution across views $C_{\text{MV}}(p)$ exceeds the pruning threshold $\delta_{\text{prune}}$. In such cases, the transmittance is reset to a lower value to restore the gradient flow and allow optimization to continue. Experimental results demonstrate that MV-Prune effectively suppresses floaters. It achieves up to 60% compression of redundant Gaussians while maintaining comparable rendering quality.

### 4.3 MULTI-VIEW CONSISTENCY

To enhance geometric consistency, we follow principles similar to MVS algorithms in adopting photometric constraints across neighboring patches. For each reference pixel $\mathbf{p}_r$ within the non-distractor region $V$ defined by $Mask_{sam}$, we compute the corresponding homography matrix and map the $11 \times 11$ patch $P_r$ around $\mathbf{p}_r$ to the corresponding region $P_n$ in a neighboring view. The consistency between these patches is measured using the normalized cross-correlation $\text{NCC}(\cdot)$ (Yoo & Han, 2009) coefficient. We thus enforce consistency between $P_r$ and $P_n$ using the following loss term:

$$\boldsymbol{L}_{mv} = \frac{1}{V} \sum_{\boldsymbol{p}_r \in V} (1 - \text{NCC}(\boldsymbol{I}_r(\boldsymbol{p}_r), \boldsymbol{I}_n(\boldsymbol{p}_n))). \tag{9}$$

We compute per-pixel geometric accuracy weights to get the regions with large reconstruction errors. Specifically, we reproject the neighboring pixel $\boldsymbol{p}_n$ back to the reference view using the homography matrix to obtain $\boldsymbol{p}'_n$. The reprojection error is then calculated as:

$$\boldsymbol{E}_{repro} = \frac{1}{V} \sum_{\boldsymbol{p}_r \in V} \|\boldsymbol{p}_r - \boldsymbol{H}_{rn}\boldsymbol{p}_n\|. \tag{10}$$

The weight for each pixel is defined as:

$$\boldsymbol{w}_{repro} = \frac{1}{1 + \boldsymbol{E}_{repro}}. \tag{11}$$

This weighting scheme reduces the influence of unreliable correspondences and naturally handles out-of-bounds and occlusion cases.

Unlike conventional MVS algorithms, we do not directly minimize the reprojection error. Without sufficient neighborhood context, this would impair the gradient propagation of Gaussian parameters. Instead, we rely on the color consistency to provide sufficient signals for depth optimization.

In addition to the photometric term, we introduce a surface regularization loss $L_s$ that minimizes the shortest axis of each Gaussian to encourage thin surface representations. We also use an image reconstruction loss $L_{rgb}$. The full loss function is defined as:

$$\boldsymbol{L} = L_{rgb} + \lambda_1 L_s + \lambda_2 w_{repro} L_{mv}, \tag{12}$$

where we set $\lambda_1 = 100$ for the surface loss and $\lambda_2 = 0.2$ for the geometric loss.

## 5 EXPERIMENTS

In this section, we present both qualitative and quantitative comparisons of our method with state-of-the-art Gaussian Splatting approaches, 2DGS (Huang et al., 2024), PGSR (Chen et al., 2024), SLS (Sabour et al., 2024), on our self-collected and public datasets.

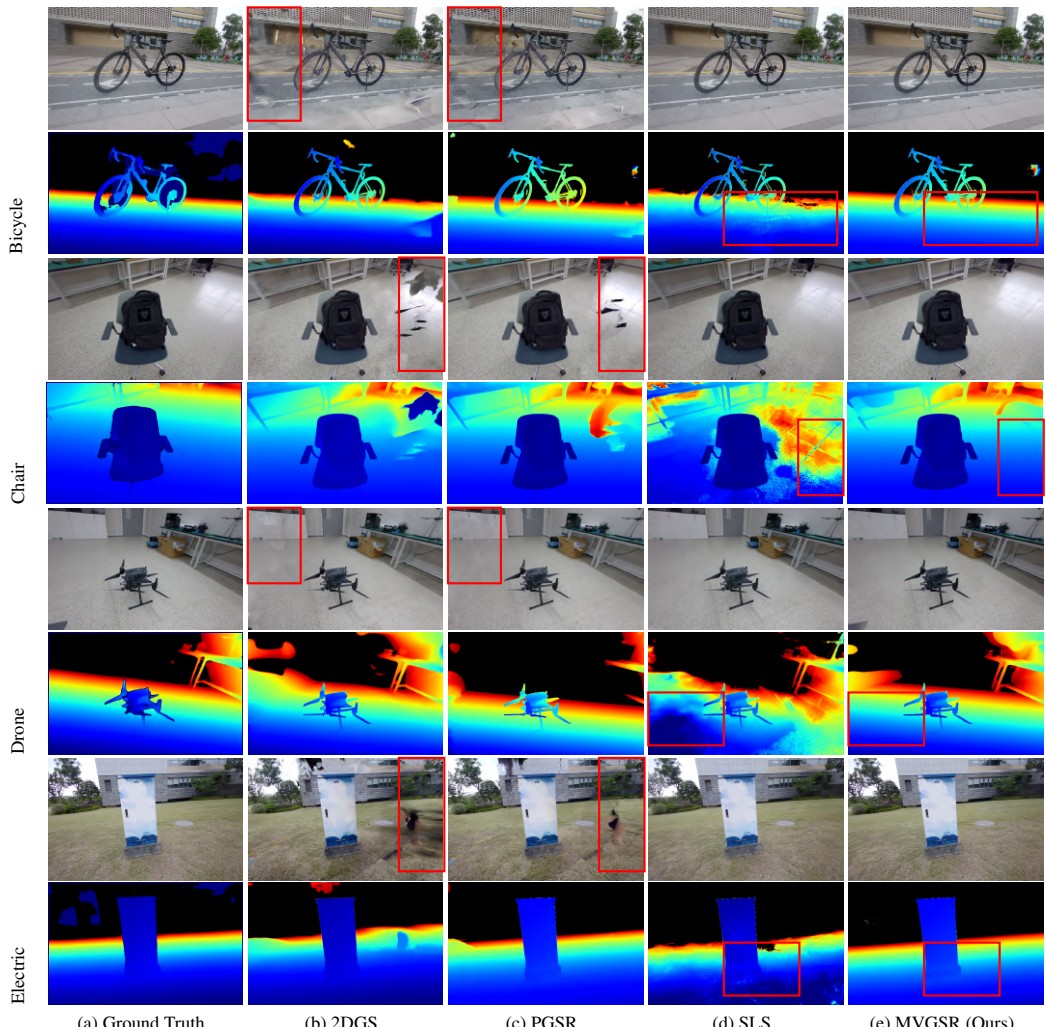

Figure 3: Comparison of RGB and depth rendering on the real-world dataset. We compare 2DGS, PGSR, SLS, and our MVGSR across four scenes. Red boxes highlight regions with distractors or challenging geometry.

## 5.1 IMPLEMENTATION

All experiments are conducted on a single NVIDIA RTX 4090 GPU. In multi-view settings, we use at most 8 closest images with an angular difference of no more than $60°$ between adjacent cameras and a maximum distance of $1.5$. The novel view rendering uses models trained for $7,000$ iterations, and view-to-view relationships are computed using Eqn. 6. For multi-view masks in Sec. 4.1, we set the cosine similarity threshold $\delta_{near}$ to 0.5. For SAM prompts, we use 20 segmentation points and 1 exclusion point. A segmentation result is considered valid if at least 10 candidates receive more than 2 votes, and the refined masks are then used for re-training. Surface reconstruction reaches high quality after $30,000$ iterations with these masks. In the multi-view pruning stage, the transmittance contribution is set to 0.5, and we retain only views with contributions above the threshold of 8.

## 5.2 DATASETS AND METRICS

**Datasets.** We evaluate reconstruction and rendering on both real and synthetic datasets. The real dataset is collected with a binocular stereo camera (Orbbec Gemini 2 XL, range 0.4-10 m) and includes four scenes (chair, drone, bicycle, electric), each with ∼100 static and ∼100 distractor images at $1280 \times 800$ resolution with RGB-depth pairs. The synthetic datasets, DTU-Robust and TnT-

Table 1: Reconstruction and rendering results on our collected dataset. SSIM ↑ , PSNR ↑ and $\delta(25\%)$ ↑ are higher the better; LPIPS ↓ and MRE ↓ are lower the better. **Bold** indicates the best results.

| Method | Bicycle | | | | | Chair | | | | | Drone | | | | | Electric | | | | |
|---|---|---|---|---|---|---|---|---|---|---|---|---|---|---|---|---|---|---|---|---|
| | SSIM | PSNR | LPIPS | $\delta(25\%)$ | MRE | SSIM | PSNR | LPIPS | $\delta(25\%)$ | MRE | SSIM | PSNR | LPIPS | $\delta(25\%)$ | MRE | SSIM | PSNR | LPIPS | $\delta(25\%)$ | MRE |
| 2DGS | 0.680 | 19.26 | 0.346 | 25.40% | 0.139 | 0.724 | 21.38 | 0.337 | 34.90% | 0.518 | 0.796 | 21.74 | 0.300 | 38.20% | 0.179 | 0.631 | 19.86 | 0.342 | 12.20% | 0.475 |
| PGSR | 0.662 | 19.91 | 0.295 | 40.10% | 0.103 | 0.883 | 22.41 | 0.146 | 43.30% | 0.365 | 0.784 | 21.48 | 0.243 | 50.50% | 0.160 | 0.729 | 21.42 | 0.211 | 33.91% | 0.222 |
| SLS | 0.767 | 22.75 | 0.198 | 16.50% | 0.157 | 0.853 | 27.13 | 0.216 | 25.50% | 0.678 | 0.894 | 25.77 | 0.183 | 19.30% | 0.395 | 0.728 | 24.55 | 0.300 | 16.80% | 0.340 |
| MVGSR | **0.796** | **23.85** | **0.170** | **49.10%** | **0.079** | **0.906** | **27.41** | **0.121** | **51.50%** | **0.307** | **0.927** | **27.61** | **0.129** | **62.80%** | **0.102** | **0.887** | **26.03** | **0.120** | **34.82%** | **0.182** |

Table 2: Quantitative results of reconstruction performance on the DTU-Robust dataset. **Bold** indicates the best results.

| | Scan | 24 | 37 | 40 | 55 | 63 | 65 | 69 | 83 | 97 | 105 | 106 | 110 | 114 | 118 | 122 | Avg. |
|---|---|---|---|---|---|---|---|---|---|---|---|---|---|---|---|---|---|
| CD↓ | PGSR | 0.53 | 1.00 | 0.51 | 0.36 | 1.14 | 0.61 | 0.49 | 0.91 | 0.62 | 0.59 | 0.47 | 0.46 | 0.32 | 0.37 | 0.35 | 0.61 |
| | 2DGS | 0.52 | 0.86 | 0.60 | 0.45 | 1.12 | 0.99 | 0.82 | 1.37 | 1.19 | 0.69 | 0.71 | 0.70 | 0.41 | 0.70 | 0.56 | 0.77 |
| | SLS | 1.82 | 2.31 | 1.17 | 2.46 | 1.95 | 1.03 | 2.28 | 1.64 | 2.12 | 1.76 | 1.38 | 2.05 | 1.53 | 1.89 | 1.21 | 1.76 |
| | MVGSR | **0.34** | **0.51** | **0.30** | **0.30** | **0.44** | **0.52** | **0.45** | **0.63** | **0.59** | **0.56** | **0.35** | **0.38** | **0.29** | **0.34** | **0.35** | **0.42** |
| PSNR↑ | PGSR | 31.34 | 26.72 | 29.88 | 31.81 | 32.66 | 31.27 | 30.94 | 32.02 | 30.37 | 32.76 | 34.94 | 33.58 | 32.16 | 36.41 | 35.71 | 32.15 |
| | 2DGS | 30.31 | 28.73 | 27.87 | 26.65 | 33.52 | 33.66 | 31.09 | 31.23 | 33.22 | 30.99 | 30.89 | 32.31 | 32.16 | 33.58 | 35.87 | 31.47 |
| | SLS | **36.85** | **30.20** | **35.75** | 32.80 | **38.15** | 33.95 | 31.50 | 39.95 | **35.60** | 37.45 | 34.60 | **36.25** | **38.40** | **39.80** | 33.80 | **35.80** |
| | MVGSR | 33.06 | 29.22 | 32.52 | **34.29** | 37.01 | **34.11** | **32.86** | 40.90 | 34.37 | **38.02** | **37.96** | 35.81 | 33.21 | 39.40 | **39.82** | 35.58 |

Robust, are constructed by injecting distractors into DTU (Jensen et al., 2014) and TnT (Knapitsch et al., 2017) sequences, covering 15 object-centric and 6 large-scale outdoor scenes with ground-truth 3D models. Scene details are provided in Appendix A.1.

**Metrics.** Rendering quality is measured by SSIM, PSNR, and LPIPS, evaluating structural fidelity and perceptual realism. Geometry is assessed using the depth accuracy ratio $\delta(25\%)$ and Mean Relative Error (MRE). For surface reconstruction, we report Chamfer Distance (CD) on DTU-Robust and F1 score on TnT-Robust, following standard practice. CD quantifies point cloud alignment, while F1 balances precision and recall under a distance threshold.

## 5.3 COMPARISON IN DEPTH AND APPEARANCE RENDERING

Tab. 1 reports reconstruction and rendering results on the four collected scenes. Baselines such as 2DGS and PGSR show degraded performance, with lower PSNR/SSIM and higher LPIPS/MRE, reflecting reduced robustness under distractors (e.g., PGSR fails on *Bicycle* and *Chair* due to noisy regions). In contrast, MVGSR consistently ranks best or second-best across all metrics, significantly improving SSIM/PSNR and reducing LPIPS/MRE. Notably, it achieves the highest $\delta(25\%)$ (62.8% on Drone), highlighting superior depth consistency. These results demonstrate the robustness and generalization of MVGSR in both indoor and outdoor settings.

Fig. 3 further shows that 2DGS and PGSR suffer from artifacts (e.g., chair edges, electric scene), while MVGSR suppresses such errors and yields smoother, more consistent surfaces aligned with the ground truth. Although SLS attains good rendering quality, it generates incorrect geometry on repetitive textures (e.g., ground, grass), indicating that robust reconstruction without depth constraints can overfit distractors.

## 5.4 COMPARISON IN SURFACE RECONSTRUCTION

We evaluate the reconstruction capability of our method against distractor-free algorithms SLS (Sabour et al., 2024), and the surface reconstruction algorithm PGSR (Chen et al., 2024) using the DTU-Robust dataset. On DTU-Robust, our method achieves a mean CD score of 0.42, surpassing 2DGS and PGSR by 76.45% and 60.31% on average as listed in Tab. 2.

We evaluate rendering performance on the DTU-Robust dataset against PGSR and 2DGS. As shown in Tab. 2, our method consistently achieves higher PSNR across all scenes, verifying stable performance under random settings. On scan83, it surpasses 2DGS and PGSR by 9.67 dB and 8.88 dB, respectively, and reaches an average of 35.58 dB, clearly outperforming baselines. Qualitative comparisons in Fig. 4 show that our method removes distractors and distortions more effectively,

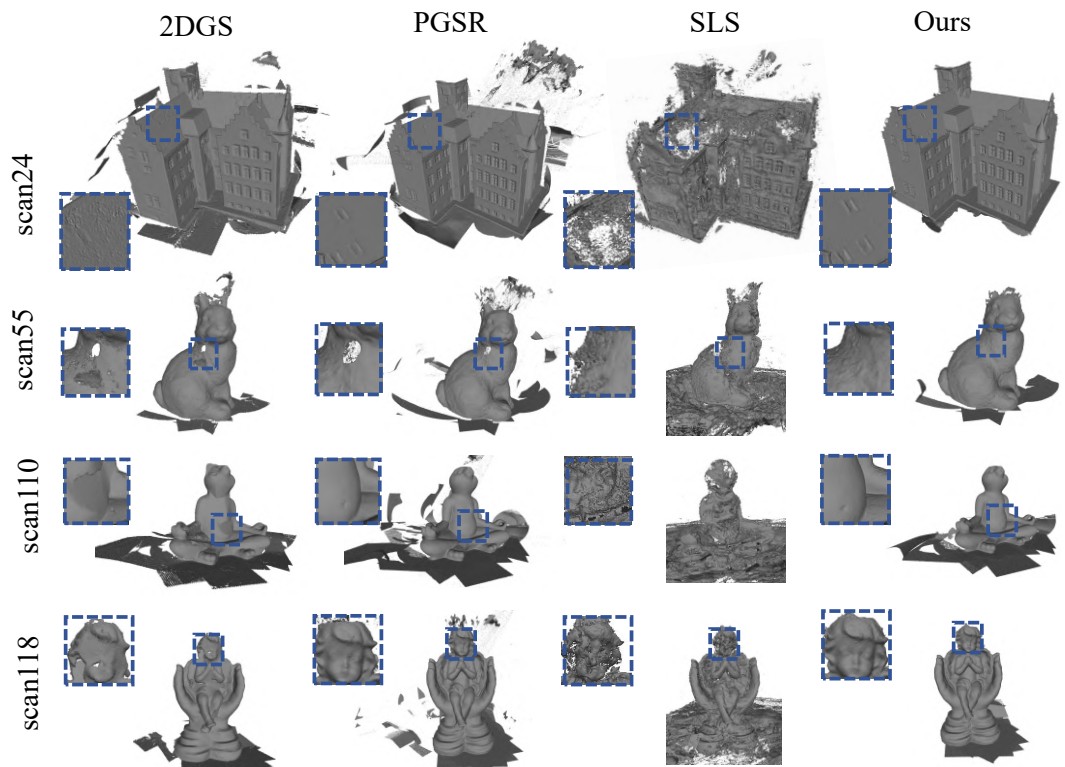

Figure 4: Qualitative comparison of surface reconstruction results on the DTU-Robust dataset. Compared with 2DGS and PGSR, our method produces cleaner geometry and sharper details. The results demonstrate improved robustness to distractors and higher-fidelity surface reconstruction.

producing reconstructions closest to the ground truth. This demonstrates the strength of our pruning strategy in alias removal. Refer to the appendix for additional results.

## 5.5 ABLATION STUDY

To demonstrate the contribution of each component, ablation experiments are conducted on DTU-Robust scan24 in Tab. 3. The multi-view Loss $L_{mv}$ significantly improves surface reconstruction quality by reducing the Chamfer Distance from 1.63 to 0.37, despite a slight decrease in rendering performance. Mask and MV-Prune effectively reduce artifacts, ultimately achieving high-fidelity rendering with superior quality preservation. Also, we compare the runtime and resource consumption of different algorithms on Scan24 of the DTU-Robust dataset. Refer to the appendix for more details.

Table 3: Ablation study of the components.

| Mask_mv | Mask_sam | $L_{mv}$ | MV-Prune | PSNR | CD |
|---------|----------|----------|----------|-------|------|
| ✓ | ✓ | | | 32.20 | 1.63 |
| ✓ | | ✓ | | 30.01 | 0.56 |
| ✓ | ✓ | ✓ | | 30.75 | 0.37 |
| ✓ | ✓ | ✓ | ✓ | 33.06 | 0.34 |

## 6 DISCUSSION AND CONCLUSION

We presented a robust surface reconstruction framework based on lightweight 3D Gaussian Splatting for dynamic and cluttered environments. Instead of assuming static scenes or relying on heavy MLP-based uncertainty modeling, our method exploits multi-view feature consistency for early distractor masking, introduces Gaussian pruning to suppress floating artifacts, and adopts a multi-view consistency loss to enhance structural and appearance fidelity. Experiments across diverse scenarios show that our approach delivers competitive geometric accuracy and photorealistic rendering while remaining robust to dynamic objects and transient distractors.

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

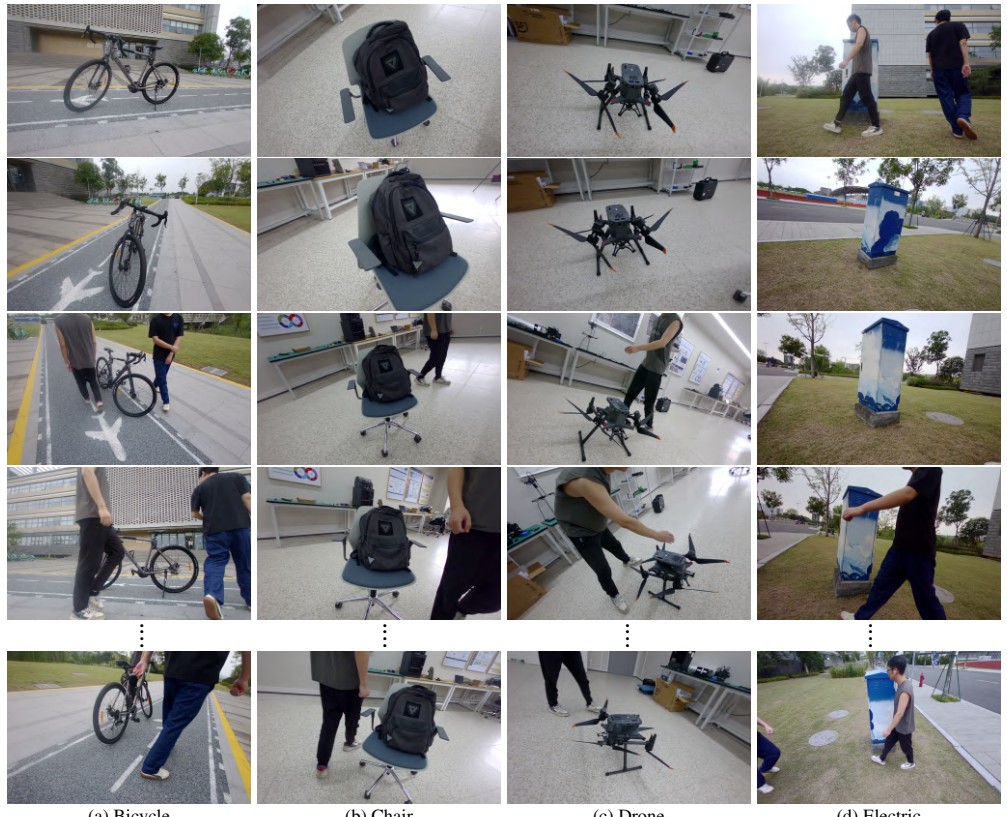

|   |   |   |   |
|---|---|---|---|
| (a) Bicycle | (b) Chair | (c) Drone | (d) Electric |

Figure 5: Overview of the real-world dataset. Sample training images are shown for four representative categories: (a) Bicycle, (b) Chair, (c) Drone, and (d) Electric. Each category contains paired scenes with and without distractors, enabling pixel-aligned comparisons.

# A    APPENDIX

## A.1    DATASET DETAILS

**Real-world Datasets.**    We evaluate reconstruction and rendering performance on real-world indoor and outdoor scenes captured with an Orbbec Gemini 2 XL stereo camera (operating range: 0.4-10 m). The dataset includes four scenes: two indoor (chair, drone) and two outdoor (bicycle, electric). For each scene, we collect about 100 images of the static target and 100 images with distractors, all at a resolution of $1280 \times 800$ pixels, with both RGB and depth information provided (Fig. 5).

**DTU-Robust and TnT-Robust.**    As shown in Fig. 6, in DTU-Robust, random geometric shapes (squares, circles, triangles) of varying sizes are superimposed on training images. TnT-Robust enhances real-world fidelity by injecting distractors (pedestrians, vehicles, animals) from DAVIS 2017 (Pont-Tuset et al., 2017) into static scenes. Both extended datasets enable comprehensive robustness testing under distractors while preserving original reconstruction benchmarks.

## A.2    QUALITATIVE AND QUANTITATIVE RESULTS IN APPEARANCE AND DEPTH RENDERING

Fig. 7 shows qualitative RGB rendering comparisons on the DTU-Robust dataset. Baseline methods, such as 2DGS and PGSR, introduce visible artifacts and distractors (e.g., color blotches and inconsistent textures), while SLS reduces some noise but produces blurry details. In contrast, our MVGSR generates cleaner and sharper renderings that closely resemble the ground truth across different scenes, demonstrating strong robustness against distractors and improved fidelity in both texture and lighting.

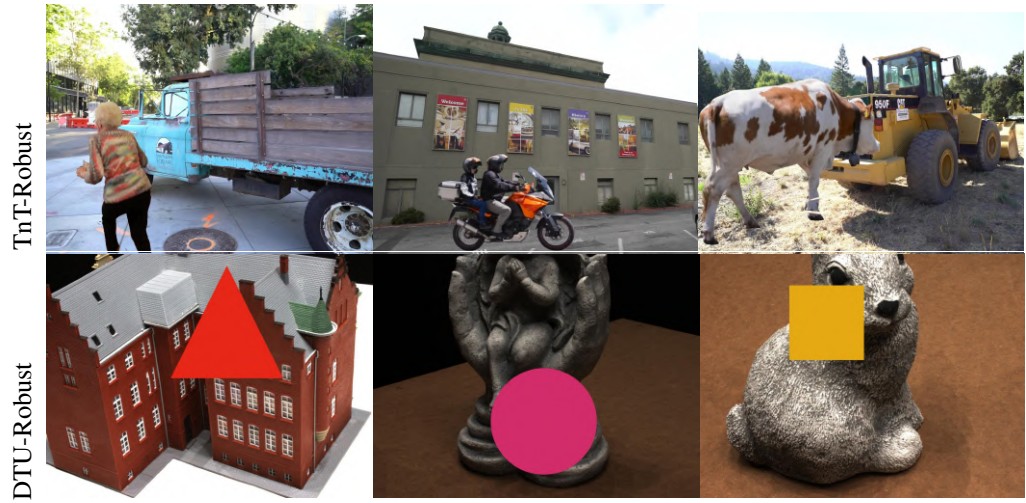

Figure 6: Overview of the synthetic datasets. TnT-Robust and DTU-Robust consist of 6 large-scale outdoor scenarios and 15 object-centric scenes, respectively. Benefiting from the original TnT and DTU datasets, each scenario in our synthetic datasets is accompanied by a ground-truth 3D surface model.

Fig. 8 presents qualitative comparisons of RGB and depth rendering on the TnT-Robust dataset. The input images (column a) contain distractors that challenge reconstruction quality. Baseline methods such as PGSR and NeRF-on-the-Go (Ren et al., 2024) suffer from severe artifacts in both RGB appearance and depth maps, while SLS achieves relatively robust rendering but still produces inconsistent geometry around distractor regions. In contrast, our MVGSR (column b) effectively removes distractor influence, yielding sharper RGB reconstructions and smoother, more accurate depth predictions that are better aligned with the true scene structure.

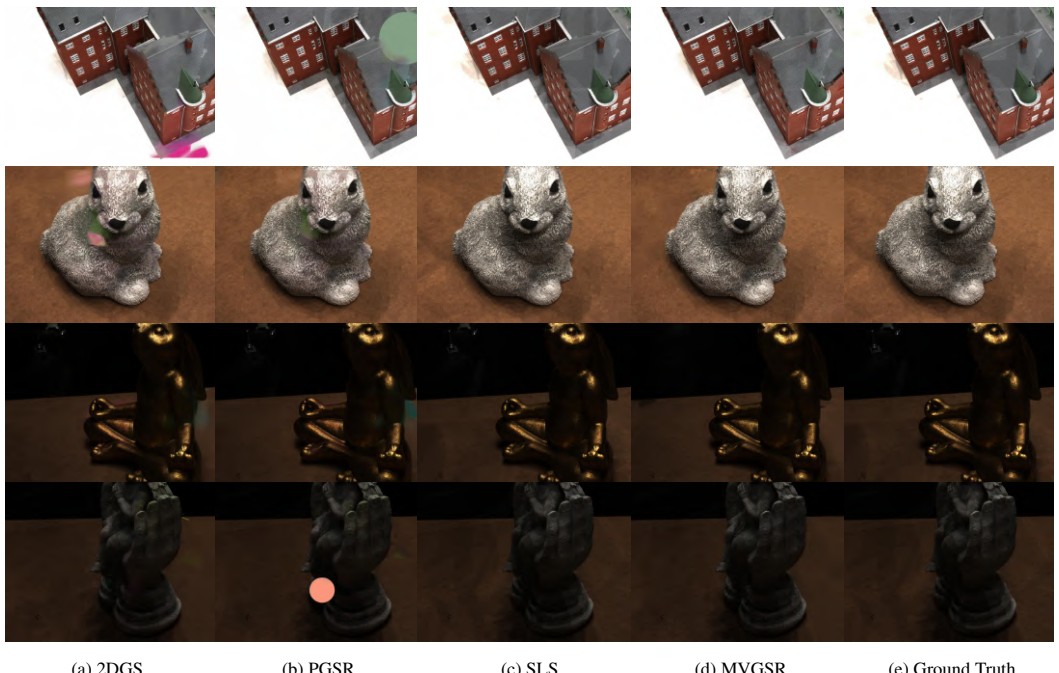

| (a) 2DGS | (b) PGSR | (c) SLS | (d) MVGSR | (e) Ground Truth |

Figure 7: Comparison of RGB rendering on the DTU-Robust dataset.

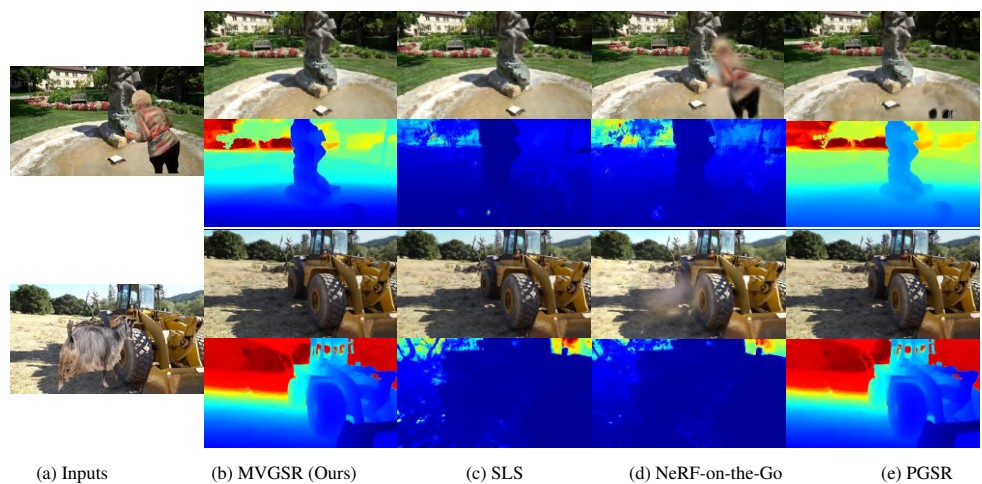

(a) Inputs     (b) MVGSR (Ours)     (c) SLS     (d) NeRF-on-the-Go     (e) PGSR

Figure 8: Comparison of RGB and depth rendering on the TnT-Robust dataset.

Table 4: Quantitative results of reconstruction performance on the TnT-Robust dataset.

|  | Sequence | Truck | Caterpillar | Barn | Meetingroom | Ignatius | Courthouse | Avg. |
|---|---|---|---|---|---|---|---|---|
| | PGSR | 0.57 | 0.37 | 0.57 | 0.30 | 0.64 | 0.18 | 0.44 |
| F1↑ | SLS | 0.48 | 0.30 | 0.43 | 0.24 | 0.57 | 0.15 | 0.36 |
| | NeRF-on-the-Go | 0.37 | 0.22 | 0.28 | 0.17 | 0.49 | 0.11 | 0.27 |
| | MVGSR | 0.60 | 0.42 | 0.59 | 0.34 | 0.73 | 0.18 | 0.48 |

Tab. 4 reports the quantitative reconstruction results on the TnT-Robust dataset using the F1 score. MVGSR achieves the best overall performance with an average F1 of 0.48, outperforming PGSR (0.44), SLS (0.36), and NeRF-on-the-Go (0.27). In particular, MVGSR shows consistent improvements across most sequences, such as Truck (0.60 vs. 0.57 PGSR) and Ignatius (0.73 vs. 0.64 PGSR). These results demonstrate that MVGSR delivers more reliable surface reconstruction under distractor-heavy scenarios compared to existing baselines.

### A.3 RUNTIME OF THE SYSTEM

Tab. 5 presents the runtime breakdown of MVGSR on DTU-Robust Scan24. The overall runtime is approximately 40 minutes, with the majority attributed to Refine Train (2029.0s). Coarse Train initialization establishes the static scene in 123.2s, followed by Render Static (19.8s). Subsequent Feature Extract (41.8s) processes raw images and rendered features, leading to Contrast Mask generation (91.7s) for multi-view comparison and Prompt SAM operations (89.9s) for refined masking. These results indicate that MVGSR remains computationally efficient while incorporating additional masking and pruning modules.

### A.4 LIMITATIONS AND FUTURE WORK

In this section, we further introduce the limitations of the proposed method. On the DTU-Robust dataset, as listed in Tab. 2, MVGSR exhibits slightly reduced rendering performance compared to SLS on several scans (e.g., Scan24, Scan40, and Scan114). The degradation is localized to regions with complex, view-dependent shadow variations, such as the camera shadow on the roof in Scan24. Because such shadows are strongly view-dependent, thin Gaussian primitives used in this paper are less expressive than Gaussian ellipsoids for capturing subtle appearance changes. Moreover, although shadows remain semantically consistent across viewpoints, MVGSR still struggles to obtain complete masks for these regions.

As shown in Fig. 9, MVGSR underperforms SLS in edge details (upper red box) and complex shadow areas (lower red box). This occurs because MVGSR adopts thin-plate Gaussians to enhance surface quality, resulting in poor fitting performance at sharp edges. While the multi-view consistency

Table 5: The Runtime metrics on DTU-Robust Scan24.

| MVGSR | Coarse Train | Render Static | Feature Extract | Mask_mv | Mask_sam | Refine Train | total |
|---|---|---|---|---|---|---|---|
| Time | 123.2s | 19.8s | 41.8s | 91.7s | 89.9s | 2029.0s | 39.93min |

constraint also improves the smoothness of the surface reconstruction, it sacrifices rendering details that exhibit color inconsistencies across different viewpoints.

As shown in the Fig. 10, principal component analysis (PCA) is employed here to reduce 384-dimensional DINO features to 3 dimensions for visualization. For visualizing DINO semantic features, they exhibit difficulty in distinguishing subtle shadow variations but perform well in handling occluded objects. In the algorithmic workflow, the absence of initial prompt points prevents subsequent SAM segmentation from further extracting masks for shadow variations.

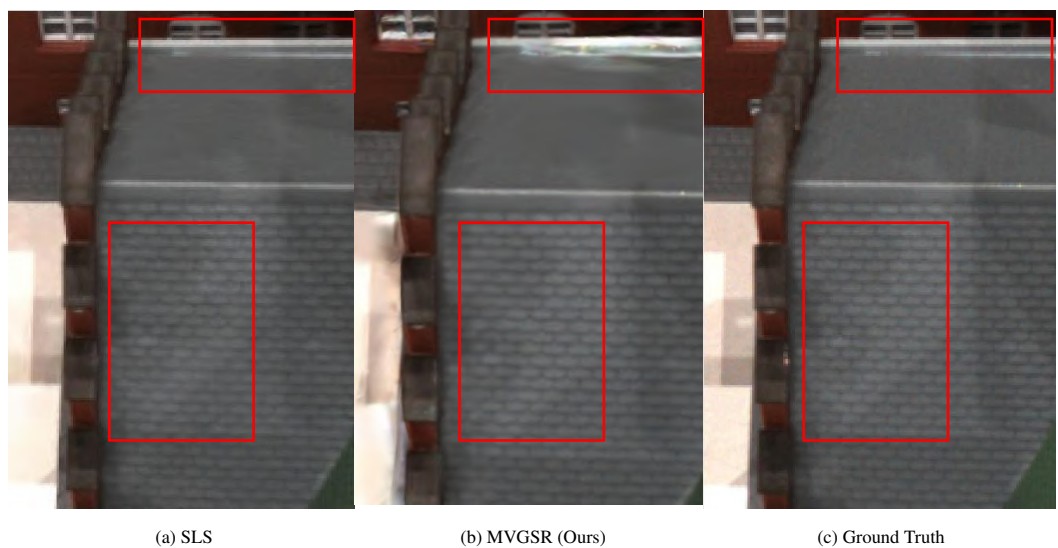

|      (a) SLS      |      (b) MVGSR (Ours)      |      (c) Ground Truth      |

Figure 9: Compare the detailed differences between SLS and MVGSR on the DTU-Robust dataset Scan24. The rendering performance of MVGSR in handling complex shadows and edge details is affected by the thin Gaussian representation, resulting in a decline in the quality of appearance rendering.

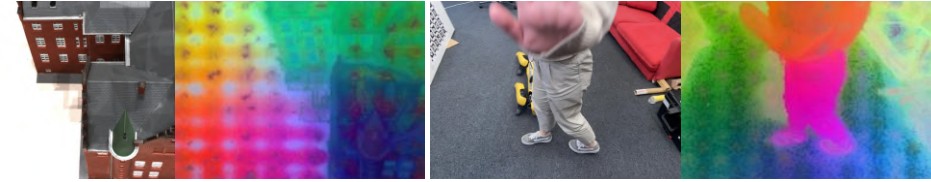

Figure 10: Visualization of DINO Semantic Features. While the features struggle to capture subtle shadow variations, they demonstrate robust recognition capabilities against distractors.

In future work, we will explore multi-scale feature pyramids and lighting-sensitive low-level features to better capture fine-grained distractors and improve robustness to lighting variations. Incorporating hierarchical features will allow the model to handle distractors of different scales more effectively, while sensitivity to illumination cues will enhance reconstruction fidelity under challenging lighting conditions such as shadows, highlights, or dynamic illumination changes. Together, these improvements are expected to further strengthen the adaptability of our framework to diverse real-world scenarios.

