# OpenReview forum: "MVGSR: Multi-View Consistency Gaussian Splatting for Robust Surface Reconstruction"
_ICLR.cc/2026/Conference — Submitted to ICLR 2026_

### Official Review · Reviewer_jLVC · 2025-10-29

**Soundness:** 2
**Presentation:** 3
**Contribution:** 2
**Rating:** 2
**Confidence:** 5

**Summary:**

This paper introduces MVGSR, a 3D Gaussian Splatting (3DGS) framework for 3D surface reconstruction that effectively handles dynamic objects and transient distractors in real-world scenes. The method’s core contributions are threefold: 1) Distractor Masking Strategy: A coarse 3DGS model is first trained, and DINOv2 features from reference and neighboring views are compared. Feature inconsistencies generate an initial distractor mask, which is refined using SAM. 2) MV-Prune Mechanism: A multi-view pruning strategy evaluates each Gaussian’s cumulative transmittance contribution across views to remove floating artifacts. 3) Photometric Consistency Loss: A multi-view NCC-based loss enforces structural and color consistency in non-distractor regions.

**Strengths:**

1. Using inter-view DINO feature consistency to detect interfering objects, rather than relying on traditional uncertainty modeling or photometric residuals, is a new perspective.
2. The paper contributes a new dataset specifically designed to evaluate the robustness of interference, which is valuable to the community.
3. The paper is easy to understand.

**Weaknesses:**

1. It has limited novelty. The photometric consistency loss derived from MVS has been adopted by many existing works. While the use of DINO feature consistency is interesting, similar concepts have already been explored in prior studies, which employ MVS network features instead of DINO features.
2. The ablation results in Table 3 demonstrate a substantial improvement in geometric accuracy when $Mask_{sam}$ is added to $Mask_{mv}$. This suggests that SAM is not merely refining boundaries but is actually responsible for the primary segmentation. Consequently, the contribution of $Mask_{mv}$ appears to function mainly as a coarse prompt for SAM.
3. SLS serves as a strong reconstruction baseline specifically designed for 3DGS. However, both the experimental results (e.g., PSNR values in Table 2) and the authors’ own statements show that MVGSR often underperforms SLS in key rendering quality metrics. Although the method achieves better geometric accuracy, it performs worse in rendering. This trade-off is insufficiently discussed, and the authors fail to justify why such a compromise is acceptable or beneficial.
4. Why are surface reconstruction results on the Tanks & Temples dataset not presented? Showing results only on DTU is insufficient for current surface reconstruction research, as DTU is considered too simple to fully demonstrate a method’s effectiveness.
5. In summary, this paper appears to pursue innovation for its own sake. It combines relatively simple techniques while presenting them as a solution to a new problem, yet it contributes little genuine originality.

**Questions:**

1. Regarding the role of SAM: The ablation results in Table 3 show that SAM yields a substantial improvement in CD, strongly indicating that it does more than merely refine boundaries. Could you provide a quantitative evaluation of $Mask_{mv}$ (before applying SAM), such as its IoU with the ground-truth distractor mask? Is it possible that the DINO feature comparison primarily serves to generate noisy cues for SAM, which is in fact performing the actual segmentation?
2. It claims that MV-Prune outperforms the standard 3DGS pruning strategy. However, standard 3DGS also performs pruning based on opacity. Could you provide a more direct ablation study comparing your MV-Prune with the standard 3DGS pruning method, while keeping all other factors (mask and loss) identical?
3. The masking strategy is based on comparing the real image $I_r$ with an image $I_n$ rendered from a coarse model trained on noisy data. Could you elaborate on why this comparison is valid? If the 7k-iteration model is already heavily corrupted by noise, making $I_n$ an inaccurate representation of the static scene, wouldn’t the feature distance $Distance(p_r, p_n)$ become unreliable?

---

### Official Review · Reviewer_NXV9 · 2025-10-30

**Soundness:** 3
**Presentation:** 2
**Contribution:** 2
**Rating:** 4
**Confidence:** 4

**Summary:**

MVGSR targets robust surface reconstruction with 3D Gaussian Splatting in scenes containing dynamic or transient distractors. It (1) flags distractors early via a multi-view feature-consistency mask (DINOv2 + SAM), (2) removes or resets spurious Gaussians using MV-Prune based on multi-view contributions, and (3) enforces a patch-wise multi-view consistency loss via per-pixel homographies. On distractor-augmented DTU/Tanks&Temples and small real RGB-D sets, it delivers state-of-the-art geometry with competitive rendering quality, with trade-offs including reliance on large vision priors and occasional edge/shadow fidelity limits.

**Strengths:**

1. Well-motivated problem & clear insight. Identifying distractors via semantic-geometry inconsistency across views is intuitive and avoids early residual-based confusion that leaks gradients into background.
2. Practical pruning strategy. MV-Prune’s alpha-driven multi-view contribution addresses floaters and compresses the Gaussian set while keeping quality, avoiding color-gradient size bias.
3. Consistent, competitive results. Strong geometry on DTU-Robust and solid PSNR/SSIM/LPIPS trends across real scenes.

**Weaknesses:**

1. Dependence on heavy priors. The pipeline leans on DINOv2 + SAM; accuracy and runtime (and domain shift) may hinge on these external models and prompt choices.
2. Appearance trade-offs. The shift toward “thin surfels” improves geometry but can reduce fidelity at sharp edges and complex shadows; SLS occasionally wins PSNR, noted by the authors.
3. Ablation scope. While component ablations exist, there’s limited exploration of alternatives (e.g., other feature backbones, no-SAM setting, different patch sizes) and of MV-Prune failure modes.

**Questions:**

1. Ablating the priors. What is the performance without SAM (mask from DINO only), or with alternative segmenters? Likewise, how do different feature encoders (e.g., CLIP, DINOv1) affect the masks?
2. Persistent distractors. The mask keeps a pixel if ≥2 neighbors agree it’s a distractor. How does the method behave when a moving object persists across many views (e.g., a parked car that later moves)? Can static content be mis-masked?

---

### Official Review · Reviewer_mFE7 · 2025-10-31

**Soundness:** 3
**Presentation:** 3
**Contribution:** 3
**Rating:** 6
**Confidence:** 4

**Summary:**

This paper addresses robust 3D reconstruction from multi-view images containing transient distractors . The authors propose MVGSR, which leverages multi-view feature consistency (via DINOv2) to detect distractors early in training, refines masks using SAM with explicit prompting, and introduces a view-contribution-based pruning strategy to remove floating artifacts. The method is evaluated on synthetic datasets and self-collected real RGB-D scenes. Results demonstrate improvements in geometric accuracy (Chamfer Distance) and rendering quality (PSNR/SSIM) over 2DGS, PGSR, and SLS baselines.

**Strengths:**

1. Addresses practical limitation of 3DGS failing under dynamic scenes with a well-motivated multi-view consistency approach using DINOv2 features and voting mechanism (≥2 views) that avoids confusing distractors with fine details.

2. Evaluated on three datasets against multiple baselines (2DGS, PGSR, SLS, NeRF-on-the-Go) with consistent improvements across metrics.

3. Provides detailed runtime breakdown, honestly discusses limitations with visual examples (thin Gaussians struggle with shadows and edges in Fig. 9-10)

**Weaknesses:**

1. Training/engineering overhead remains non-trivial (SAM refinement + extra training stage). Even though the runtime is disclosed and reasonable, adoption adds pipeline complexity.

2. However, failure cases in complex shadows & thin edges persist (explicitly admitted).

**Questions:**

1. Why are some works, e.g., HybridGS not compared despite being directly relevant cited works?

2. What is the quantitative mask quality (IoU vs. manual annotations) and failure rate analysis showing when multi-view consistency or SAM fails?

---

### Official Review · Reviewer_3yz5 · 2025-11-01

**Soundness:** 2
**Presentation:** 3
**Contribution:** 2
**Rating:** 4
**Confidence:** 4

**Summary:**

This paper focuses on robust surface reconstruction in scenes with dynamic objects and distractors. To localize the dynamic objects and distractors in input images, the method first uses GS to build an initial 3D model. Then, the method renders images, depths and normal from this model for a reference image and its neighboring images. By warping the reference image into the neighboring image with rendered geometry and comparing the warped image with the rendered neighboring image, the method determines the initial mask for dynamic objects. Based on the initial mask, the method further uses SAM to refine the mask. Moreover, the method introduces multi-view contribution pruning and enhanced multi-view photometric consistency. The experiments on the collected real-world dataset, DTU and TnT show that the method can achieve robust reconstruction results in scenes with dynamic objects and distractors.

**Strengths:**

1. The method leverages multi-view inconsistency to determine initial masks for dynamic objects and distractors and then leverage SAM to refine masks. It is simple and effective.
2. The experiments on real-world dataset, DTU and TnT show that the proposed method reconstruct robust surface meshes in scenes with dynamic objects and distractors.
3. It is easy to read and understand the work.

**Weaknesses:**

1. It is better to choose and construct strong baseline methods. There are some more robust Gaussian splatting methods in the wild, such as HybridGS [1], DroneSplat [2] and DeSplat [3]. These methods do not focus on surface reconstruction and can achieve robust NVS in scenes with dynamic objects and distractors. It is better to combine these methods with 2DGS or PGSR to build strong baseline methods first to demonstrate the challenges in these special scenes. Moreover, it is better to compare with these methods in terms of NVS performance on .
[1] Jingyu Lin, et al. "HybridGS: Decoupling Transients and Statics with 2D and 3D Gaussian Splatting". In CVPR 2025.
[2] Jiadong Tang et al. "DroneSplat: 3D Gaussian Splatting for Robust 3D Reconstruction from In-the-Wild Drone Imagery". In CVPR 2025.
[3] Yihao Wang et al. "DeSplat: Decomposed Gaussian Splatting for Distractor-Free Rendering". In CVPR 2025.
2. The surface reconstruction performance of the proposed method on DTU is puzzling. The reconstruction performance of PGSR on clean DTU is 0.52. However, the performance of the proposed method on contaminated DTU, 0.42, is much better than 0.52. Can you explain this? If so, why does the method degrade on the contaminated TnT?
3. The ablation study is only conducted on one scan of DTU. It is better to conduct on the entire DTU to show the effectiveness of different components.

**Questions:**

1. For the weakness 1, if the SOTA methods are first used to obtian clean images by replacing dynamic objects and distractors with clean renderings, then PGSR is used based on the clean images. How about the performance? It is better to compare with HybridGS [1], DroneSplat [2] and DeSplat [3] on NeRFOn-the-go and RobustNeRF datasets.
2. For the multi-view consistency, can you verify the effectiveness of Eq. (11)?
3. Without the multi-view contribution pruning, would the performance improve if the masked regions are excluded in the positional gradient computation and loss functions?

---

### Meta-Review · Area_Chair_HJDw · 2026-01-03

**Summary:**

he paper is recommended for rejection due to three critical weaknesses:

Lack of Strong Baselines: The experimental design is incomplete and lacks meaningful benchmarks. The method is not compared against relevant contemporary works like HybridGS, DroneSplat, or DeSplat, which are crucial for establishing its value in challenging real-world scenarios with distractors or dynamics. This omission severely undermines the validity of its claimed contributions.

Inconsistent and Insufficient Evaluation: The reported surface reconstruction results are inconsistent (e.g., contradictory performance on DTU vs. Tanks & Temples) and inadequately validated. Key ablation studies are limited to a single DTU scan, and results on standard outdoor benchmarks like Tanks & Temples are missing, failing to demonstrate the method's general robustness.

Limited and Incremental Contribution: The core components (photometric consistency, DINO features, and reliance on SAM) are largely adaptations or recombinations of existing ideas, lacking substantial novelty. The work presents a modest trade-off (slightly better geometry for worse rendering quality) without a compelling justification for its overall benefit, appearing more like an incremental assembly of known techniques than a significant scientific advancement.

**Reviewer Concerns:**

No response is submitted by authors.

**Reviewer Scores:**

Given no response, no one would change the score.

---

### Decision · Program_Chairs · 2026-01-26

Reject